# Perspectives on the Psychological and Physiological Effects of Forest Therapy: A Systematic Review with a Meta-Analysis and Meta-Regression

Sung Ryul Shim [1,†], JinKyung Chang [2,†], JooHee Lee [3], WooJin Byeon [3], Jeongwon Lee [4] and Kyung Ju Lee [5,*]

1. Department of Health and Medical Informatics, Kyungnam University College of Health Sciences, 7, Gyeongnamdeahak-ro, Masanhappo-gu, Changwon-si 51767, Republic of Korea
2. Department of Family Medicine, Korea University Guro Hospital, 148, Gurodong-ro, Guro-gu, Seoul 08308, Republic of Korea
3. Department of Public Health, Korea University Graduate School, 145, Anam-ro, Seoungbuk-gu, Seoul 02841, Republic of Korea
4. Forest Welfare Research Center, Korea Forest Welfare Institute, 209-1, Therapy-ro, Bonghyeon-myeon, Yeongju-si 36043, Republic of Korea
5. Department of Women's Rehabilitation, National Rehabilitation Center, 58, Samgaksan-ro, Gangbuk-gu, Seoul 01022, Republic of Korea
* Correspondence: drlkj4049@korea.kr; Tel.: +82-2-901-1805
† These authors contributed equally to this work.

**Abstract:** The effects of urbanization on the health of city dwellers have spurred research on the health-promoting benefits of forest exposure, and potential health-promoting benefits of human-nature relationships. In this meta-analysis, meta-regression, and systematic review, we aimed to analyze how forest-based interventions improved overall well-being through psychological and physiological changes by examining psychological scores and biomarkers. In December 2021, systematic searches were conducted on bibliographic databases (PubMed, Embase, and Cochrane) for studies involving psychological scores and physiological indicators. Data from 17 studies with 1418 participants showed that psychological symptoms (anxiety, depression, anger, fatigue, confusion, vigor), systolic blood pressure (BP), and malondialdehyde levels significantly improved in the forest-exposed group compared to in the nonexposed group, with high heterogeneity ($I^2 = 66\%–93\%$). Well-being-related psychological symptoms (friendliness, well-being, attention deficit hyperactivity disorder, self-esteem) and physiological markers (diastolic BP and cortisol) exhibited better tendencies in the forest-exposed group, with high heterogeneity ($I^2 = 16\%–91\%$), and meta-regression showed that moderators (age, country group, number of participants, study design, female participation rate, BMI) were significantly associated with forest-related therapeutic effects. In conclusion, forest visits have health-promoting effects that reduce the incidence of stress and lifestyle-related diseases, and are positively associated with psychological and physiological health.

**Keywords:** forest therapy; health effect; psychological; physiological; stress; lifestyle

## 1. Introduction

Urbanization has been found to affect the health of city dwellers, which has spurred research on the positive effects of nature and their use in promoting public health [1]. A sedentary lifestyle, unhealthy eating habits, and unplanned urbanization accompanied by a lack of exercise are known risk factors for various chronic diseases, such as obesity, diabetes mellitus, and metabolic syndrome [2]. An urban upbringing and life are known to cause more psychological stress than a rural life [3,4]. Consequently, there is a growing body of literature on the potential benefits of human-nature relationships, and studies are being performed to investigate and discover the health-promoting benefits of forest exposure [5,6]. Research, especially when conducted from the perspective of preventive

medicine, has illustrated how exposure to greenness enhances the well-being of people visiting forests [7–9]; additionally, studies have monitored the changes in physiological and psychological parameters after exposure to forest environments [10,11].

Forest-based interventions—such as simple walks, meditation, and leisure activities performed in forests—have contributed to physiological and psychological recovery, as these interventions have been found to reduce the levels of depression, anxiety, fatigue, and stress; enhance social cooperation; induce a sense of well-being; and aid in recovering self-esteem [12–17]. This phenomenon has been reported along with supporting biological findings, including significantly lower salivary cortisol concentrations and serum malondialdehyde (MDA) levels, leading to a reduction in blood pressure [18–24].

The primary research, systematic reviews, and meta-analyses conducted on this topic so far have revealed various beneficial health effects of forest-based interventions [25–28]; however, the overall physical and mental outcomes have not been systematically reviewed by identifying moderator effects on individual covariates. Therefore, we first attempted forest therapy to determine the outcomes of forest interventions, which were then used to investigate the effectiveness of forest healing based on its physical and psychological effects. In this meta-analysis, we aimed to synthesize data obtained from randomized clinical trials on how forest-based interventions improved the general well-being of participants, as well as investigate whether the resulting psychological and physiological changes interact with each other. We attempted to include studies involving both psychological scores and biological markers, such as oxidative and hypertensive properties, to uncover the association between the psychological changes and antioxidant and antihypertensive properties of forest therapy.

## 2. Materials and Methods

This systematic review and meta-analysis was registered at PROSPERO (CRD42021283846) and was conducted in accordance with the preferred reporting items for systematic reviews and meta-analyses (PRISMA) statement [29] and the Meta-analysis of Observational Studies in Epidemiology reporting guideline (Supplementary Table S2).

### 2.1. Data Sources and Literature Searches

Through December 2021, we conducted comprehensive literature searches in the PubMed, Embase, and Cochrane databases using terms from the Medical Subject Headings (MeSH) list and text keywords; our search strategy was as follows: population of interest (i.e., healthy volunteers or patients diagnosed with underlying diseases), interventions (exposure to forest environments), and outcomes (mental health measurement scores and biomarkers) (Supplementary Table S1). Our search focus was papers with an obvious forest-based intervention and a comparable control group. We conducted the initial search without restrictions on language or the type of study; subsequently, we manually searched for additional studies in clinical trial databases and reference lists.

### 2.2. Eligibility Criteria for Study Selection

The following inclusion criteria were used for study selection: (1) study population comprising healthy volunteers or patients diagnosed with underlying diseases, such as mental disorders, fibromyalgia, and hypertension (HTN); (2) interventions involving exposure to forest environments; (3) comparisons with an unexposed group; and (4) study outcomes comprising the standardized mean differences (SMD) in the Profile of Mood States (POMS; tension-anxiety, depression, anger-hostility, fatigue-inertia, confusion-bewilderment, and vigor-activity) scores, friendliness, well-being, attention deficit hyperactivity disorder (ADHD), self-esteem, systolic blood pressure (SBP), diastolic blood pressure (DBP), malondialdehyde (MDA), and cortisol concentrations between the forest-exposed and unexposed groups.

### 2.3. Data Extraction and Measurement Outcomes

Two independent investigators analyzed the titles and abstracts based on the inclusion and exclusion criteria, and examined the full-text articles using the same criteria. After selecting the articles, the authors independently extracted data using a data extraction form. In case of discrepancies between the two authors' opinions, they reached an agreement through group discussions. All the investigators evaluated and discussed the final articles selected for inclusion. To prevent overlapping data and to maintain meta-integrity, the references and data from each included study were carefully crosschecked.

### 2.4. Data Analysis and Statistical Analysis

All variables with improvements were recorded as continuous data. SMDs (Hedges' g) with 95% confidence intervals (CI) and the DerSimonian and Laird random-effects model were used to adequately analyze the overall SMDs, and because each study had a different follow-up period, the outcome value recorded at the end of the study was used [3].

Cochran's Q test and the $I^2$ statistic were used to assess the proportion of total heterogeneity due to within-study and between-study variations. In the meta-regression analysis, we used a restricted maximum likelihood estimator of the variance to analyze the effects of potential moderators (e.g., number of patients, age, female participation rate, body mass index (BMI), country, study design, crossover study, underlying disease, forest program education, forest therapist, and forest program duration). A two-sided $p$-value $\leq 0.05$, or 95% CI not containing the null value (ratio = 1), were considered statistically significant. This analysis was performed with R software version 4.0.3 (R Foundation for Statistical Computing, Vienna, Austria) [3].

To assess methodological quality, the risk of bias and methodological quality were evaluated in duplicate using the Cochrane Collaboration tool. We assessed six parameters: (1) random sequence generation; (2) allocation concealment; (3) blinding of participants, personnel, and outcome assessors; (4) incomplete outcome data; (5) selective outcome reporting; and (6) other sources of bias. All domains were classified as having high, low, or unclear risk of bias.

Potential publication bias was assessed using a funnel plot in which standard errors were used as a measure of research sample sizes and ratio assessments of treatment effects. We considered that the overall effect size of the studies would be symmetrically distributed in the absence of publication bias. To statistically evaluate publication bias, we used the Begg and Mazumdar rank correlation and Egger's linear regression tests.

## 3. Results

### 3.1. Study Selection

During the initial search, we identified 861 articles from different electronic databases (PubMed, n = 267; Cochrane, n = 9; Embase, n = 582) and manual searches (n = 3). Sixty-one studies that contained overlapping data or appeared in more than one database were excluded. After screening the titles and abstracts, 682 studies were eliminated because they were unrelated to the topic in question. Of these, 81 studies were excluded for the following reasons: no target diseases (n = 25), unclear forest therapy (n = 16), no outcome value (n = 8), commentary or letter (n = 30), and others (n = 2). Finally, 17 studies met our selection criteria for the qualitative and quantitative syntheses (Figure 1).

A systematic review of the 17 studies was conducted to assess the differences and subject descriptions in detail (Table 1). The total number of participants was 1418, including both healthy volunteers and patients diagnosed with underlying diseases; the average age was 11.8 to 70.5 years. The studies were conducted in different countries across the globe, and based on the meta-analysis, the study designs included randomized controlled trial, cross-sectional, and crossover studies.

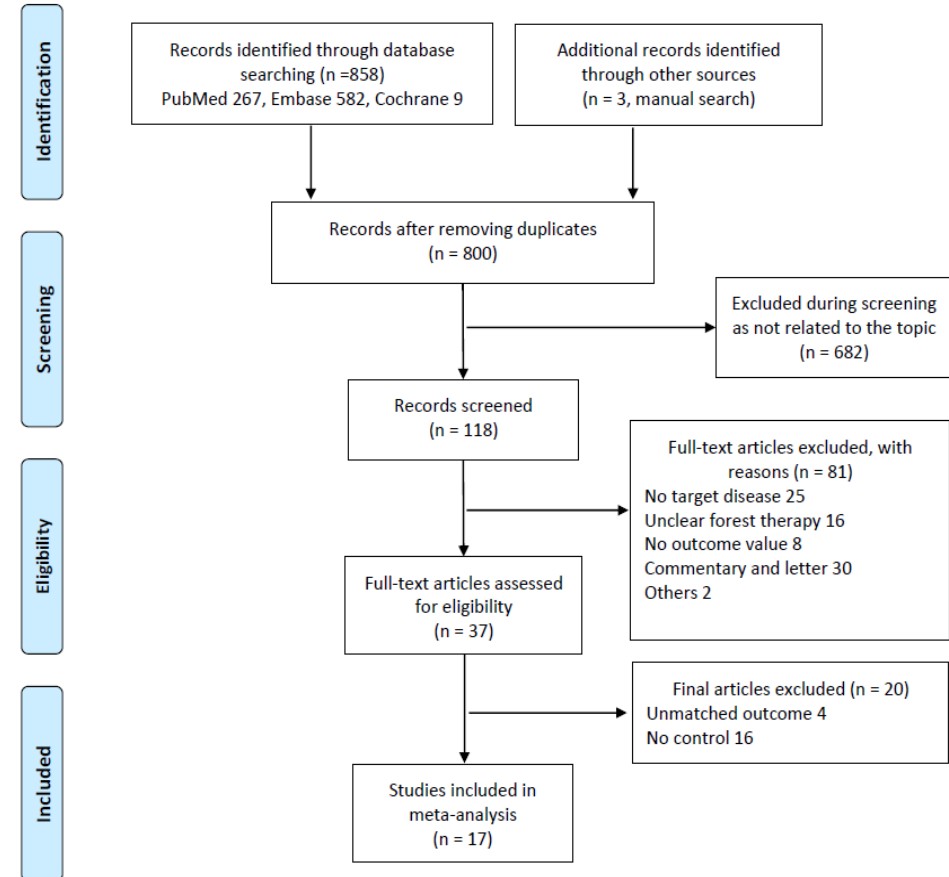

**Figure 1.** PRISMA flow diagram of the search strategy and study selection. PRISMA, preferred reporting items for systematic reviews and meta-analyses.

**Table 1.** Characteristics of the included studies on the psychological and physiological effects of forest therapy.

| References | Country | Study Design | Number of Participants (Mean Age) | Population (Number in Control Group, Number in Experimental Group) | Total Time of Intervention (min) | Intervention | | Outcome Measurements |
|---|---|---|---|---|---|---|---|---|
| | | | | | | Experimental Group | Control Group | |
| Ameli et al., 2021 [18] | USA | Crossover | 12 (35) | Healthy male volunteers from a US military facility | 20 | Walking in a forested woodland | Walking around a university campus | Level of anxiety (DT), level of concentration (MAAS) |
| Zeng et al., 2020 [22] | China | RCT | 120 (21) | Healthy university students (30, 90) | 45 | Walking in bamboo forests in Sichuan Province | Walking in downtown Chengdu | Heart rate, DBP, SBP |
| Kim et al., 2020 [10] | Korea | RCT | 38 (22) | Healthy university students (19, 19) | 90 | Walking and stretching in forests | Daily routine activities | POMS * |
| Takayama et al., 2019 [15] | Japan | RCT | 46 (21) | Healthy male university students (23, 23) | 15 | Walking along forest roads | Walking around downtown major traffic routes or near the main train station | POMS, sense of wellbeing (SVS) |
| Kobayashi et al., 2019 [24] | Japan | Crossover | 74 (22.4) | Healthy male university students | 15 | Walking in one of the seven forests | Walking near city centers or railway stations | Salivary cortisol concentration |

**Table 1.** *Cont.*

| References | Country | Study Design | Number of Participants (Mean Age) | Population (Number in Control Group, Number in Experimental Group) | Total Time of Intervention (min) | Intervention | | Outcome Measurements |
|---|---|---|---|---|---|---|---|---|
| | | | | | | Experimental Group | Control Group | |
| Wallner et al., 2018 [16] | Austria | Crossover | 60 (16.6) | Healthy adolescents from three different high schools in Vienna | 60 | Walking in forests | Walking in urban parks | Level of concentration (d2-R test) |
| Bang et al., 2018 [12] | Korea | RCT | 52 (11.78) | Healthy elementary school students (28, 24) | 600 | Walking and playing sports in forests | Studying at a community center | Sense of wellbeing, level of self-esteem (Rosenberg Self-Esteem Scale), depression (K-CDI), sociability, level of concentration (K-CWAS) |
| Lee et al., 2018 [27] | Korea | RCT | 71 (55.73) | Females volunteers diagnosed with or without metabolic syndrome (38, 33) | 120 | Walking, playing folk games, and meditating in a wild forest | Walking, playing folk games, and meditating in a tended park | POMS, SBP, DBP, MDA level |
| Stigsdotter et al., 2017 [11] | Denmark | Crossover | 51 (NA) | Healthy female university students in Copenhagen | 15 | Walking in the Danish Health Forest | Walking in downtown Copenhagen | POMS, SBP, DBP |
| Bang et al., 2017 [29] | Korea | RCT | 99 (24.31) | Graduate and undergraduate students in Seoul (48, 51) | 240 | Walking in forests | Daily routine activities | Depression (BDI), sociability (degree of social relation), SBP, DBP |
| Sonntag-Ostrom et al., 2015 [13] | Sweden | RCT | 86 (44.6) | Participants diagnosed with exhaustion disorder (46, 40) | 2640 | Relaxation exercises and meditation in forests | Daily routine activities | Level of fatigue (Checklist Individual Strength Questionnaire), Level of self-esteem (Self-concept Questionnaire), depression and level of anxiety (HADS) |
| Lopez-Pousa et al., 2015 [17] | Spain | RCT | 30 (62.3) | Participants diagnosed with fibromyalgia (16, 14) | 360 | Walking in a mature forest | Walking in a young forest | Depression (FIQR), level of anxiety (STAI), number of days with a sense of well-being |
| Lee et al., 2014 [21] | Korea | RCT | 62 (70.47) | Healthy female volunteers (19, 43) | 60 | Walking in a Pyunback tree forest in Mokpo city | Walking in the downtown area of Mokpo city | SBP, DBP |
| Sung et al., 2012 [23] | Korea | RCT | 56 (64.5) | Participants diagnosed with stage 1 hypertension (28, 28) | 1440 | Walking and meditating in two forests (Hoeungseong and Saneum) | Walking around nearby urban areas | SBP, DBP, salivary cortisol concentration |
| Mao et al., 2012 [28] | China | RCT | 20 (20.79) | Healthy male university students (10, 10) | 180 | Walking in the broad-evergreen Wuchao Mountain forest in Hangzhou | Walking in Hangzhou city areas | MDA level |
| Kim et al., 2009 [14] | Korea | RCT | 42 (43.78) | Participants diagnosed with MDD (19 hospitalized, 23 outpatients) | 720 | Mindfulness meditation at Hong-Reung arboretum | Mindfulness meditation in a room in Seoul Paik Hospital | Salivary cortisol concentration, sense of wellbeing (SFHSQ) |

**Table 1.** *Cont.*

| References | Country | Study Design | Number of Participants (Mean Age) | Population (Number in Control Group, Number in Experimental Group) | Total Time of Intervention (min) | Intervention | | Outcome Measurements |
|---|---|---|---|---|---|---|---|---|
| | | | | | | Experimental Group | Control Group | |
| Morita et al., 2007 [8] | Japan | Crossover | 498 (56.2) | 244 healthy male and 254 healthy female volunteers | 150 | Walking and exercising in Tokyo University Forest | Walking and exercising at home | Level of hostility, depression, boredom, sociability, sense of wellbeing, and liveliness (MMF-SF), level of anxiety (STAI) |

BDI, Beck Depression Inventory; DBP, diastolic blood pressure; DT, Distress Thermometer; FIQR, Revised Fibromyalgia Impact Questionnaire; HADS, Hospital Anxiety and Depression Scale; HR, heart rate; K-CDI, Korean Version of the Children's Depression Inventory; K-CWAS, Korean Version of Conners-Wells Adolescents Self-Report Scales; MAAS, Mindful Attention Awareness Scale-State Version; MDA, malondialdehyde; MDD, major depressive disorder; MMF-SF, Multiple Mood Scale-Short Form; POMS, Profile of Mood States; SBP, systolic blood pressure; SFHSQ, Short Form Health Survey Questionnaire; STAI, Spanish Version of State-Trait Anxiety Inventory; SVS, Subjective Vitality Scale. * POMS measures anxiety, depression, hostility, vigor, fatigue, and confusion.

*3.2. Quality Assessment*

The selected studies were critically appraised by the authors using the risk of bias measure of the Cochrane Collaboration tool. Of the 17 studies, nine described randomization methods, and only three conducted allocation concealment. The overall quality of the studies was evaluated as low because the exposure involved was an environmental exposure that could not be controlled in a laboratory.

*3.3. Outcomes*

In the meta-analysis, the pooled overall SMDs in the POMS scores for psychological symptoms significantly improved in the forest-exposed group compared to the nonexposed group (anxiety: SMD, $-0.650$; 95% CI, $-1.092$ to $-0.208$; depression: SMD, $-0.650$; 95% CI, $-1.092$ to $-0.208$; anger: SMD, $-0.555$; 95% CI, $-0.992$ to $-0.118$; fatigue: SMD, $-0.631$; 95% CI, $-1.211$ to $-0.051$; confusion: SMD, $-1.057$; 95% CI, $-1.922$ to $-0.192$; and vigor: SMD, $0.710$; 95% CI, $0.072$ to $1.348$). Cochran's Q test revealed high heterogeneity ($I^2$ = 66%–93%) (Figure 2).

In the meta-analysis, the pooled overall SMDs in well-being-related psychological symptoms were not significantly improved in the forest-exposed group compared to the nonexposed group (friendliness: SMD, $-0.025$; 95% CI, $-0.196$ to $0.145$; wellbeing: SMD, $0.322$; 95% CI, $-0.046$ to $0.690$; ADHD: SMD, $0.360$; 95% CI, $-0.738$ to $1.457$; self-esteem: SMD, $0.271$; 95% CI, $-0.601$ to $1.143$). Cochran's Q test showed high heterogeneity ($I^2$ = 16%–91%) (Figure 3).

Regarding changes in the analyzed biomarkers, the pooled overall SMDs in SBP and MDA levels significantly improved in the forest−exposed group compared to the nonexposed group (SBP: SMD, $-0.932$; 95% CI, $-1.690$ to $-0.175$; MDA: SMD, $-0.664$; 95% CI, $-1.089$ to $-0.240$). There were no significant improvements in the pooled overall SMDs in DBP and cortisol concentrations in the forest−exposed group compared to the nonexposed group (DBP: SMD, $-0.434$; 95% CI, $-0.901$ to $0.033$; cortisol: SMD, $-1.023$; 95% CI, $-2.105$ to $0.059$) (Figure 4).

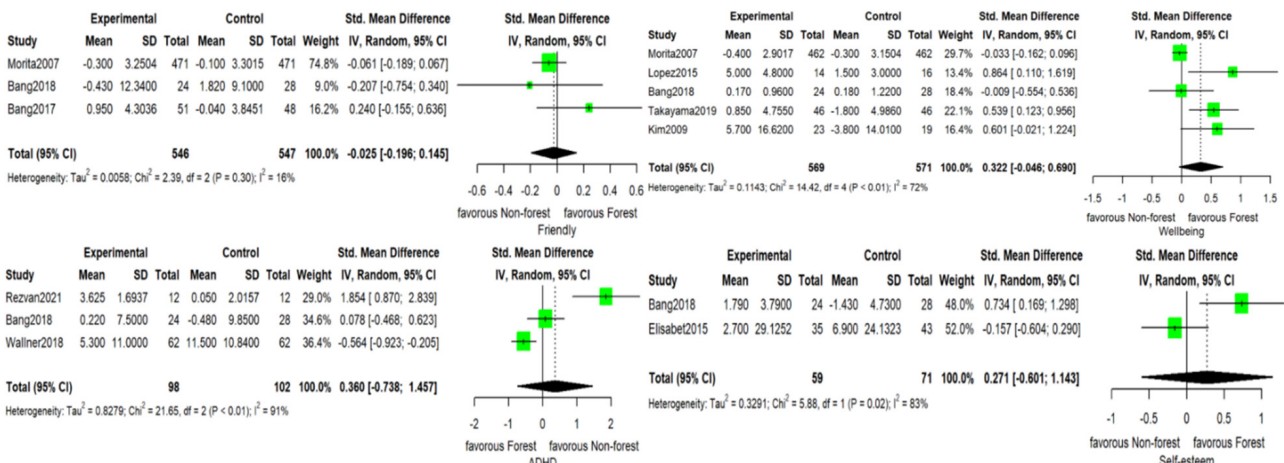

**Figure 2.** Forest plot of the results of a random-effects meta-analysis shown as mean differences in the POMS scores for psychological symptoms with 95% CIs. POMS, Profile of Mood States; CI, confidence interval; SD, standard deviation.

**Figure 3.** Forest plot of the results of a random-effects meta-analysis shown as mean differences in well-being-related psychological symptoms with 95% CIs. CI, confidence interval; SD, standard deviation; ADHD, attention-deficit/hyperactivity disorder.

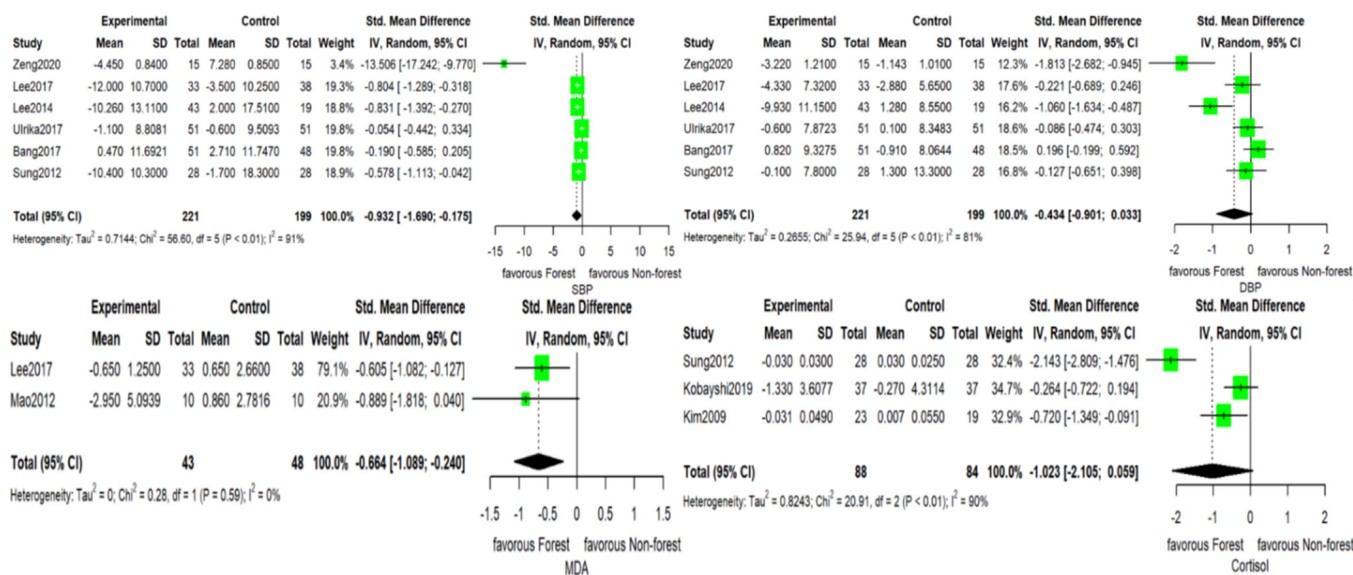

**Figure 4.** Forest plot of the results of a random-effects meta-analysis shown as mean differences in physiological biomarkers with 95% CIs. CI, confidence interval; SD, standard deviation; SBP, systolic blood pressure; MDA, malondialdehyde; DBP, diastolic blood pressure.

### 3.4. Effect Size Modifiers

Tables 2 and 3 present the subgroup analysis results, and overview of the meta-regression analysis results. In the meta-regression analysis, we found that age, country group, number of patients, study design, female participation rate, and BMI were significantly associated with forest therapeutic effects. The older age group showed more severe anxiety symptoms (age: regression coefficient of SMD, 0.044; 95% CI, 0.001 to 0.087). In studies with a small number of subjects (a large difference between 498 and 20), SBP was lower in the forest therapy group (regression coefficient of SMD, −0.138; 95% CI, −0.257 to 0.019); in particular, the higher ratio of females, the higher the SBP in relation to the forest therapy effect (regression coefficient of SMD, 14.172; 95% CI, 0.465 to 27.878). In addition, the higher the BMI, the weaker the forest therapy effect (regression coefficient of SMD, 2.290; 95% CI, 0.298 to 4.282). Several studies were conducted in Asian countries (Asia: SMD, −0.048; 95% CI, −0.761 to −0.215; others: SMD, 0.008; 95% CI, −0.404 to 0.421), and other moderators—such as underlying disease, forest program education, involvement of a forest therapist, and forest program duration—did not have significant effects.

**Table 2.** Results of the meta-regression analysis of the effects of forest therapy.

| | | Anxiety | | | | | Depression | | | | | Well-Being | | | |
|---|---|---|---|---|---|---|---|---|---|---|---|---|---|---|---|
| Variables | k | SMD | 95% CI | | p | k | SMD | 95% CI | | p | k | SMD | 95% CI | | p |
| **Total no. of patients** | 8 | 0.002 [a] | −0.002 | 0.007 | 0.331 | 9 | 0.000 [a] | −0.002 | 0.003 | 0.740 | 5 | −0.001 [a] | −0.002 | 0.000 | 0.038 [†] |
| **Age** | 7 | 0.044 [a] | 0.001 | 0.087 | 0.043 [†] | 8 | 0.009 [a] | −0.009 | 0.027 | 0.312 | 5 | 0.003 [a] | −0.018 | 0.024 | 0.762 |
| **Female participation rate** | 7 | 1.629 [a] | −0.385 | 3.643 | 0.113 | 8 | 0.438 [a] | −0.541 | 1.417 | 0.381 | 4 | −0.191 [a] | −1.495 | 1.113 | 0.774 |
| **BMI** | - | - | - | - | - | 3 | 0.045 [a] | −0.080 | 0.170 | 0.479 | - | - | - | - | - |
| **Country** | | | | | 0.719 | | | | | 0.049 [†] | | | | | 0.220 |
| Asia | 4 | −0.779 | −1.493 | −0.066 | | 6 | −0.488 | −0.761 | −0.215 | | 4 | 0.229 | −0.128 | 0.587 | |
| Others | 4 | −0.590 | −1.331 | 0.152 | | 3 | 0.008 | −0.404 | 0.421 | | 1 | 0.865 | −0.085 | 1.814 | |
| **Study design** | | | | | 0.163 | | | | | 0.532 | | | | | 0.060 |
| Cross-sectional | 4 | −1.083 | −1.842 | −0.324 | | 3 | −0.473 | −0.936 | −0.011 | | 1 | −0.033 | −0.424 | 0.358 | |
| RCT | 4 | −0.333 | −1.062 | 0.396 | | 6 | −0.290 | −0.631 | 0.051 | | 4 | 0.462 | 0.126 | 0.798 | |

**Table 2.** *Cont.*

| | | Anxiety | | | | | Depression | | | | | Well-Being | | | |
|---|---|---|---|---|---|---|---|---|---|---|---|---|---|---|---|
| Variables | k | SMD | 95% CI | | p | k | SMD | 95% CI | | p | k | SMD | 95% CI | | p |
| **Crossover study** | | | | | 0.751 | | | | | 0.257 | | | | | 0.060 |
| Yes | 3 | −0.578 | −1.388 | 0.232 | | 2 | −0.109 | −0.605 | 0.387 | | 1 | −0.033 | −0.424 | 0.358 | |
| No | 5 | −0.745 | −1.379 | −0.110 | | 7 | −0.449 | −0.764 | −0.135 | | 4 | 0.462 | 0.126 | 0.798 | |
| **Underlying disease** | | | | | 0.351 | | | | | 0.264 | | | | | 0.092 |
| Yes (patients) | 2 | −0.274 | −1.244 | 0.696 | | 2 | −0.053 | −0.616 | 0.509 | | 2 | 0.716 | 0.133 | 1.298 | |
| No (healthy) | 6 | −0.807 | −1.367 | 0.248 | | 7 | −0.409 | −0.677 | 0.509 | | 3 | 0.139 | −0.196 | 0.473 | |
| **Forest program education** | | | | | 0.679 | | | | | 0.124 | | | | | 0.844 |
| Yes | 3 | −0.816 | −1.630 | −0.001 | | 5 | −0.535 | −0.881 | −0.189 | | 2 | 0.282 | −0.414 | 0.978 | |
| No | 5 | −0.599 | −1.221 | 0.023 | | 4 | −0.148 | −0.500 | 0.204 | | 3 | 0.370 | −0.155 | 0.895 | |
| **Forest therapist involvement** | | | | | 0.801 | | | | | 0.240 | | | | | 0.558 |
| Yes | 4 | −0.746 | −1.461 | −0.031 | | 5 | −0.506 | −0.871 | −0.141 | | 3 | 0.444 | −0.103 | 0.991 | |
| No | 4 | −0.618 | −1.309 | 0.072 | | 4 | −0.210 | −0.543 | 0.124 | | 2 | 0.215 | −0.322 | 0.753 | |
| **Forest program time (min)** [b] | | | | | 0.691 | | | | | 0.217 | | | | | 0.437 |
| <100 | 3 | −0.880 | −1.819 | 0.060 | | 3 | −0.183 | −0.663 | 0.297 | | 1 | 0.539 | −0.222 | 1.301 | |
| 100–300 | 3 | −0.830 | −1.753 | 0.094 | | 3 | −0.720 | −1.218 | 0.223 | | 1 | −0.033 | −0.683 | 0.617 | |
| ≥300 | 2 | −0.279 | −1.417 | 0.858 | | 3 | −0.163 | −0.695 | 0.368 | | 3 | 0.440 | −0.079 | 0.959 | |

k, number of effect sizes; OP, operation; ATD, antithyroid drugs; BMI, body mass index; RCT, randomized controlled trial; CI, confidence interval; SMD, standardized mean difference; [a] regression coefficient; [b] divided by the 1st and 3rd quartile. *p*-value acquired from the meta-regression analysis using the restricted maximum likelihood. [†] *p*-value < 0.05. Meta-regression analysis for continuous variables (total no. of patients, age, female participation rate, and BMI); meta-ANOVA analysis for categorical variables (country, study design, crossover study, underlying disease, forest program education, forest therapist involvement, and forest program time).

**Table 3.** Results of the meta-regression analysis of the effects of forest therapy (continued).

| | Systolic Blood Pressure | | | | | Diastolic Blood Pressure | | | | |
|---|---|---|---|---|---|---|---|---|---|---|
| Variables | k | SMD | 95% CI | | p | k | SMD | 95% CI | | p |
| Total no. of patients | 6 | −0.138 [a] | −0.257 | −0.019 | 0.023 [†] | 6 | −0.012 [a] | −0.036 | 0.012 | 0.319 |
| Age | 5 | 0.141 [a] | −0.079 | 0.361 | 0.208 | 5 | 0.004 [a] | −0.034 | 0.043 | 0.823 |
| Female participation rate | 6 | 14.172 [a] | 0.465 | 27.878 | 0.043 [†] | 6 | 1.659 [a] | −0.699 | 4.018 | 0.168 |
| BMI | 5 | 2.290 [a] | 0.298 | 4.282 | 0.024 [†] | 5 | 0.195 [a] | −0.233 | 0.624 | 0.372 |
| Country | | | | | 0.272 | | | | | 0.531 |
| Asia | 5 | −1.284 | −2.253 | −0.314 | | 5 | −0.534 | −1.130 | 0.063 | |
| Others | 1 | −0.054 | −2.022 | 1.914 | | 1 | −0.086 | −1.353 | 1.182 | |
| Study design | | | | | 0.066 | | | | | 0.373 |
| Cross-sectional | 2 | −2.256 | −4.391 | −0.735 | | 2 | −0.810 | −1.762 | 0.143 | |
| RCT | 4 | −0.598 | −1.623 | 0.428 | | 4 | −0.287 | −0.930 | 0.355 | |
| Crossover study | | | | | 0.272 | | | | | 0.531 |
| Yes | 1 | −0.054 | −2.022 | 1.191 | | 1 | −0.086 | −1.353 | 1.182 | |
| No | 5 | −1.284 | −2.253 | −0.314 | | 5 | −0.534 | −1.130 | 0.063 | |
| Underlying disease | | | | | 0.622 | | | | | 0.584 |
| Yes (patients) | 1 | −0.578 | −2.519 | 1.364 | | 1 | −0.127 | −1.383 | 1.129 | |
| No (healthy) | 5 | −1.120 | −2.057 | −0.183 | | 5 | −0.512 | −1.077 | 0.054 | |
| Forest program education | | | | | 0.278 | | | | | 0.792 |
| Yes | 4 | −1.435 | −2.554 | −0.317 | | 4 | −0.398 | −1.046 | 0.250 | |
| No | 2 | −0.435 | −1.856 | 0.987 | | 2 | −0.546 | −1.438 | 0.346 | |

**Table 3.** *Cont.*

| Variables | | Systolic Blood Pressure | | | | | Diastolic Blood Pressure | | | |
|---|---|---|---|---|---|---|---|---|---|---|
| | k | SMD | 95% CI | | p | k | SMD | 95% CI | | p |
| Forest therapist involvement | | | | | 0.809 | | | | | 0.697 |
| Yes | 1 | −0.804 | −2.732 | 1.125 | | 1 | −0.221 | −1.487 | 1.044 | |
| No | 5 | −1.068 | −2.007 | −0.130 | | 5 | −0.498 | −1.080 | 0.084 | |
| Forest program time (min) [b] | | | | | 0.766 | | | | | 0.793 |
| <100 | 4 | −1.429 | −2.668 | −0.190 | | 4 | −0.614 | −1.361 | 0.132 | |
| 100–300 | 1 | −0.804 | −3.050 | 1.442 | | 1 | −0.221 | −1.678 | 1.236 | |
| ≥300 | 1 | −0.578 | −2.835 | 1.680 | | 1 | −0.127 | −1.603 | 1.350 | |

k, number of effect sizes; OP, operation; ATD, antithyroid drugs; BMI, body mass index; RCT, randomized controlled trial; CI, confidence interval; SMD, standardized mean difference; [a] regression coefficient; [b] divided by the 1st and 3rd quartile. *p*-value acquired from the meta-regression analysis using the restricted maximum likelihood. [†] *p*-value < 0.05. Meta-regression analysis for continuous variables (total no. of patients, age, female participation rate, and BMI); meta-ANOVA analysis for categorical variables (country, study design, crossover study, underlying disease, forest program education, forest therapist involvement, and forest program time).

### 3.5. Risk of Bias Assessment

We used the Risk of Bias Assessment tool for Non-randomized Studies (RoBANS) to assess the quality of the observational studies (Figure 5). Two studies had a high risk of bias, one had an unclear risk of bias, and the other six had a low risk of bias in the participant selection process. The overall risk of bias was considered low when the effect size was not affected in the subgroup analysis.

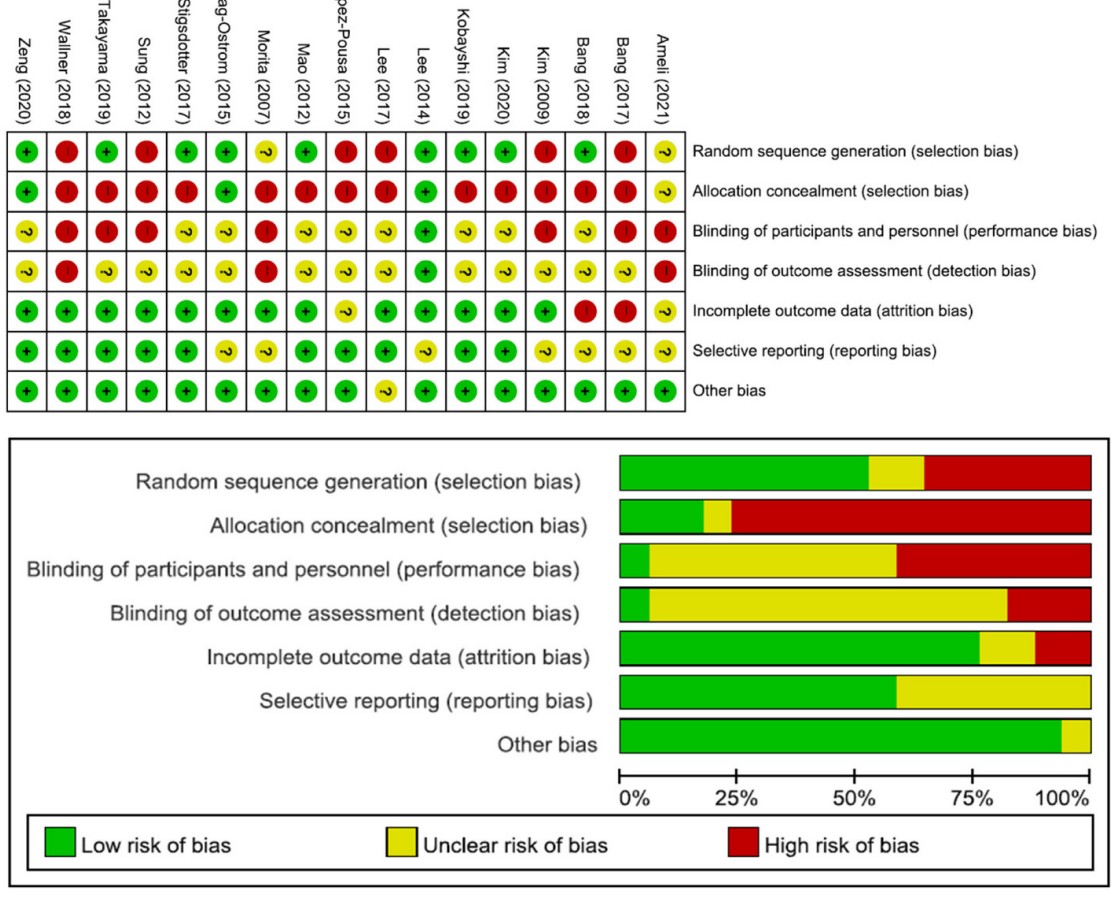

**Figure 5.** Summary of the risk of bias assessment. Refs are [8,10–15,17,18,20–22,25,30–32].

*3.6. Publication Bias and Effect Size Modifiers*

Publication bias was evaluated using the method shown in Figure 6. The funnel plots of the SMDs resulting from forest therapy appeared asymmetrical for all outcomes owing to insufficient observations. The p-values obtained through Begg and Mazumdar's correlation and Egger's regression coefficient tests suggest evidence of publication bias or small-study effects in this meta-analysis; this was not observed for the depression and well-being scores.

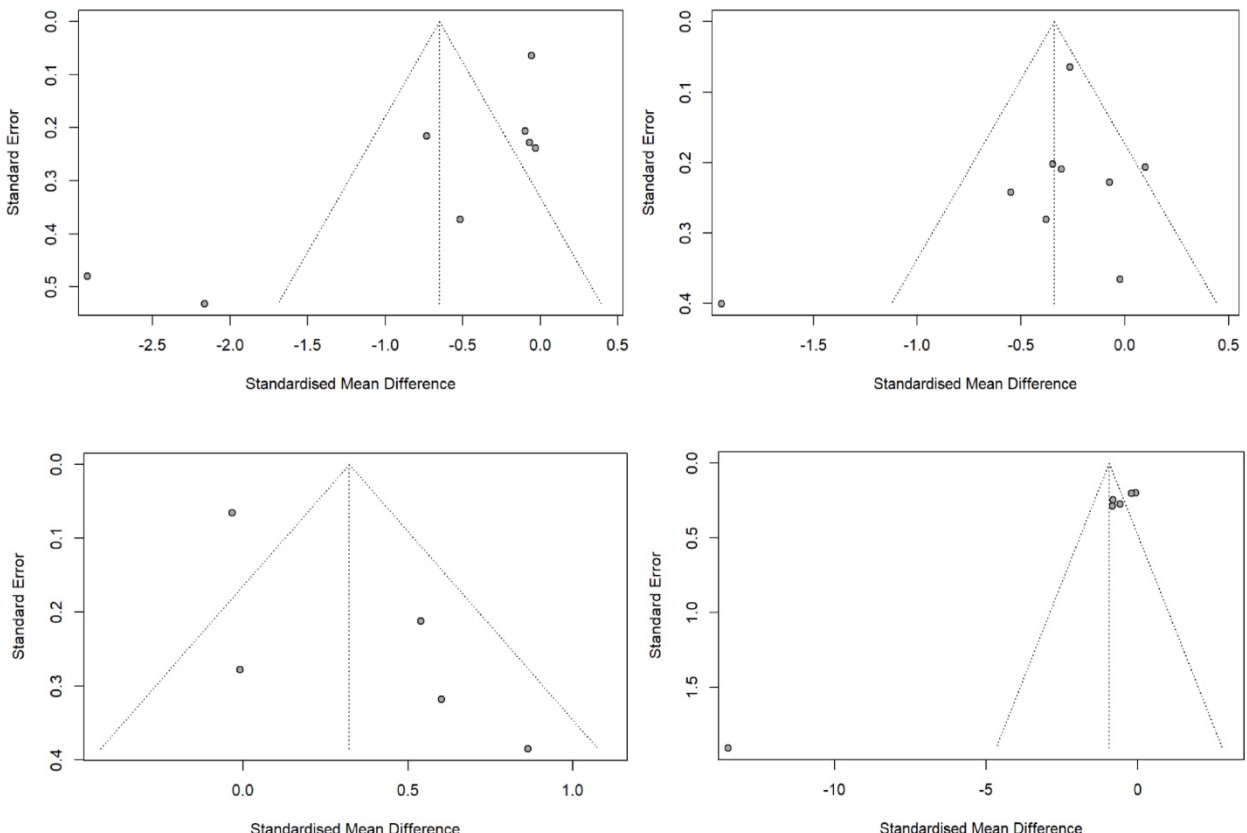

**Figure 6.** Funnel plots of the SMDs resulting from forest therapy. The SMDs appear asymmetrical for all outcomes. SMD, standardized mean difference.

## 4. Discussion

Our systematic review and meta-analysis, as well as our meta-regression analysis, demonstrated the well-known positive psychological effects and physiological changes associated with forest-based interventions. This study found that compared to the nonexposed group, psychological and well-being related symptoms exhibited better tendencies in the forest-exposed group depending on forest therapy modifiers, such as age, country group, number of participants, study design, female participation rate, and BMI.

The psychological effects of forest therapy were investigated, and significant improvements were observed in the six parameters of the POMS questionnaire—including anxiety, depression, anger, fatigue, confusion, and vigor—in the forest-exposed group. The scores measuring sociability, sense of well-being, and self-esteem improved in the forest-exposed group, but not to a significant degree when compared to those in the nonexposed group. Most studies on how exposure to forests positively affects human well-being were focused on psychological changes assessed using the POMS questionnaire [10,11,15,21].

On investigating which of the factors of forest exposure contributed to these positive psychological changes, we found that biogenic volatile organic molecules (VOCs) may be linked to effects on the nervous system. VOCs, which are also called "phytoncides," are substances emitted into the atmosphere by plants [33], and when inhaled, biogenic VOCs have

the ability to participate in the biological processes of animals and humans [34]. Preclinical studies involving animal behavioral models examined the neurological functions of biological VOCs in the central nervous system and found unexpected health-promoting effects, including the boosting of immunity and improvements in depression [35,36]. In vivo VOCs have been shown to improve sleep and alleviate pain and anxiety in mice by reducing motor activity and promoting muscle relaxation by acting as positive modulators of specific receptors [36].

In this review, SBP in the forest-exposed group was significantly lower than in the nonexposed group (Figure 3) [11,20,21,25,30,31], and some studies have reported similar effects for forest-based interventions in patients diagnosed with HTN [19,37]. The physiological mechanisms underlying lowered blood pressure can be attributed to decreased sympathetic activity, which can be measured by the levels of urinary adrenaline and/or noradrenaline and cortisol and heart rate variability (HRV) [38]. Blood pressure is known to be controlled by two parallel systems, the sympathetic and parasympathetic nervous systems, with the sympathetic system raising the blood pressure and the parasympathetic system lowering it [38,39]. Previous research on this topic revealed that activities performed in forests increased the high-frequency components of HRV and lowered the ratio of low frequency to high frequency components [40,41]. This demonstrates how forest-based interventions have calming effects by triggering the parasympathetic nervous system [24]. This implies that forest-based interventions can be used as a curative method for HTN and as a preventative method for potential HTN.

In this review, we found that cortisol, a biomarker of physiological stress, is often used to measure physiological stress levels in participants exposed to a forest environment [13,18,20,23]. The body's fight-or-flight response under physical or mental stress stimulates the sympathetic nervous system, causing a spike in cortisol or "cortisol surge", which enables the body's rapid involuntary response to dangerous and stressful situations in a state of physical danger, as well as to acute and chronic emotional awakenings [4,6,38,42–46]. By contrast, the parasympathetic nervous system releases a variety of neurohormonal substances that counteract the body's arousal state [38]. Several studies have found significantly increased parasympathetic activity in forest-exposed groups, suggesting that forests have relaxing effects [47,48]; however, while we did not find a significant decrease in cortisol levels resulting from forest-based activities in this review, we succeeded in showing a decreasing tendency in these levels (Figure 2). To demonstrate how exposure to forests lowers cortisol levels, a one-day field trip to a forest was conducted and was found to be related to lower blood cortisol and urine adrenaline levels, proving that exposure to greenness has a protective effect against stressful events at a molecular level [5,49].

We were intrigued by how the two nervous systems might have interacted in the forest-exposed groups to mediate the resulting psychological and physiological changes. The parasympathetic nervous system participates in lowering the body's blood pressure in a relaxed state with the help of biogenic VOCs, whereas the sympathetic nervous system is involved in raising the body's cortisol levels in stressful situations. Since the two nervous systems participate in regulating both blood pressure and the stress response, we hypothesized that exposure to forests lowered cortisol levels by activating the parasympathetic nervous system. This lowered cortisol level may have reduced the activity of the sympathetic nervous system, leading to lower blood pressure. Considering that diverse psychological stresses influence cortisol levels, this probable relationship might explain how the improved POMS scores were related to positive physiological changes in the forest-exposed groups.

In this meta-analysis, we found that the level of MDA, which is a biological stress marker indicative of the degree of oxidation at the cellular level, was significantly lower in the forest-exposed group than in the unexposed group (Figure 3). Some studies have reported on lower levels of MDA after forest exposure, and on a biogenic VOC called limonene, which scavenges harmful free radicals in the body due to its antioxidant properties [21,22,34–36]. Fewer free radicals would ultimately lead to cell proliferation, anti-

apoptosis of cells, and deoxyribonucleic acid (DNA) protection in human epithelial cells and fibroblasts [50,51].

The limitations of the current meta-analysis include the selection of RCTs with small sample sizes [16,22,32] and forest-based interventions, with varying protocols among the studies. The effect size was examined using 17 individual studies; however, the meta-regression analysis, which was performed to analyze detailed outcome variables, was limited to the interpretation of the meta-analysis that included up to nine studies. The inclusion of studies with small sample sizes was sufficient to obtain qualitative data, but was not sufficient for broad reasoning and the generalization of results to all age groups and environments. For example, in the modifier analysis, forest therapy resulted in a significant decrease in SBP; however, it did not affect the other analyzed variables, which might have been caused by the small to large sample sizes ranging between 20 and 498 participants. This analysis implies that an appropriate sample size must be considered along with the study method.

In some studies, subjects decided whether to participate in the intervention or control group and selected the location for the intervention based on their accessibility and preference. This selection bias implies that the personality of participants may have compromised the interpretation of our meta-analysis. This may have influenced subsequent analyses, as subjects who were motivated and induced to modify their behavior would have voluntarily participated in the experimental group and would have been more likely to benefit from forest-based activities. Furthermore, evidence regarding how forest-based interventions affect elements of sentiments—such as self-esteem, concentration, and sociability—is lacking. This prompts further research on the relationship between diverse aspects of the subjective sense of well-being and exposure to forests, especially considering the subject's personality.

Our meta-analysis showed both psychological and physiological improvements resulting from forest-based interventions, and we attempted to explain the link between the two. Research has demonstrated that exposure to various elements in the forest, such as landscapes, biogenic VOCs, smells, and sounds, contributes to physical and emotional well-being. Several European countries are interested in integrating nature into children's education [52]; diverse outdoor activities and school field trips are thus available for students to engage in outside the classroom. Some programs might be geared toward building self-esteem and confidence, while others emphasize communication and teamwork [5,52].

In recent years, there has been growing global interest in using forest environments to restore and promote health. Even brief exposure to forests during short visits is associated with psychophysical relaxation. Tables 2 and 3 show that there was a decrease in anxiety, depression, fatigue, or high blood pressure in patient groups compared to healthy control groups owing to exposure to forest environments as part of forest education, work with forest therapists, or forest programs with durations of 100–300 min.

In conclusion, this review suggests that forest visits have health-promoting effects that reduce the incidence of stress- and lifestyle-related illnesses. Additionally, it suggests that the effects are positively associated with factors such as age, country group, number of participants, study design, female participation rate, and BMI. Even though the effects of certain modifiers (forest education, forest program duration of 100–300 min, and forest therapist involvement) were not significant in the meta-analysis, we suggest considering these modifiers in future studies.

**Supplementary Materials:** The following supporting information can be downloaded at: https://www.mdpi.com/article/10.3390/f13122029/s1, Table S1: Search strategy for the review; Table S2: MOOSE checklist for meta-analyses of observational studies.

**Author Contributions:** Conceptualization, S.R.S., J.C. and K.J.L.; Methodology, S.R.S., J.C. and K.J.L.; Software, S.R.S., J.C. and K.J.L.; Validation, S.R.S., J.C. and K.J.L.; Formal Analysis, S.R.S., J.C. and K.J.L.; Investigation, S.R.S., J.C. and K.J.L.; Resources, S.R.S., J.C., J.L. (JooHee Lee), W.B., J.L. (Jeongwon Lee) and K.J.L.; Data Curation, S.R.S., J.C. and K.J.L.; Writing—Original Draft Preparation, S.R.S., J.C. and K.J.L.; Writing—Review and Editing, S.R.S., J.C., J.L. (JooHee Lee), W.B., J.L. (Jeongwon

Lee) and K.J.L.; Visualization, S.R.S., J.C., J.L. (JooHee Lee), W.B., J.L. (Jeongwon Lee) and K.J.L.; Supervision, K.J.L.; Project Administration, K.J.L.; Funding Acquisition, K.J.L. All authors have read and agreed to the published version of the manuscript.

**Funding:** This research was funded by the Korea Forestry Promotion Institute (grant number: 2021384A00-2123-0101).

**Data Availability Statement:** The datasets generated and analyzed during the current study are not publicly available because of a lack of permission but are available from the corresponding author upon reasonable request.

**Acknowledgments:** We would like to thank Backil Sung (Department of Biology, Pacific Union College) to reviewing the manuscript, and Editage for editing and reviewing this manuscript for English language.

**Conflicts of Interest:** The authors declare no conflict of interest. The funders had no role in the design of the study; in the collection, analyses, or interpretation of data; in the writing of the manuscript; or in the decision to publish the results.

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
