# Peer review of "Perspectives on the Psychological and Physiological Effects of Forest Therapy: A Systematic Review with a Meta-Analysis and Meta-Regression"

_forests, doi:10.3390/f13122029_

Round 1

Reviewer 1 Report

Dear authors I find your work of the highest quality as it succeeds in exploring the topic with a clear scientific approach.

Q1 : Line 202-212 and Table 2 should be added below, To determine whether a variable is an effect size modifier in the meta regression
(1) Significance of regression coefficient
(2) test of moderator: statistical check whether the variable act as an effect-modifier or not
(3) The degree of description of the heterogeneity of the model (R2 or How I2 reduced with the regression model or subgroup) should be presented together.
Q2 : Subgroup analysis shows that it is necessary to report the overall heterogeneity (I2) and the change in heterogeneity (I2) when divided into each subgroup
Q3 : In Table 2, it is necessary to distinguish between "Meta regression analysis" and "Meta ANOVA" (or subgroup analysis). Forwarding is not clear when reporting at once.
(1) It is necessary to present the results of each analysis of continuous variables (participants, age, femal rate, BMI) and dummy variables in two separate tables, or to add columns that are separated in the table and change the title.
Q4 : There is a risk of bias due to the small number of studies. The author needs to report the results of Eger's test in addition to the Funnel plot for publication bias. In addition, it is necessary to add related discussions to the limitations on the risk of research bias.

Author Response

Dear authors I find your work of the highest quality as it succeeds in exploring the topic with a clear scientific approach.

Q1: Line 202-212 and Table 2 should be added below, To determine whether a variable is an effect size modifier in the meta regression 

 Responses: We appreciate your valuable comments and have revised the manuscript per your suggestions. Please find our responses below.

(1) Significance of regression coefficient 

(2) Test of moderator: statistical check whether the variable act as an effect-modifier or not 

Responses: We added a footnote († P-value < 0.05”), and have already described the significant modifiers (line 202-204), p-value, and 95% CI displaying statistical significance in Table 2.

(3) The degree of description of the heterogeneity of the model (R2 or How I2 reduced with the regression model or subgroup) should be presented together.

 Responses: We described the specific heterogeneity of I2 and Cochrane’s Q statistics test, and show the exact numeric values of heterogeneity in Figures 2–4.

Q2: Subgroup analysis shows that it is necessary to report the overall heterogeneity (I2) and the change in heterogeneity (I2) when divided into each subgroup 

 Responses: We appreciate your valuable comments. As detailed information is beneficial to readers’ understanding, we described the overall heterogeneity of the main analysis for each outcome (e.g., anxiety, depression, etc.); however, in the subgroup analysis for identifying modifiers, displaying the overall and change in heterogeneity for each subgroup analysis seemed to be an excessive provision of information. Additionally, the subgroup analysis included a relatively smaller number of effect sizes than the main analysis; thus, subgroup analysis generally focused on the differences between separate groups, rather than each group’s completeness.

Q3: In Table 2, it is necessary to distinguish between "Meta regression analysis" and "Meta ANOVA" (or subgroup analysis). Forwarding is not clear when reporting at once. 

(1) It is necessary to present the results of each analysis of continuous variables (participants, age, female rate, BMI) and dummy variables in two separate tables, or to add columns that are separated in the table and change the title.

Responses: We appreciate your valuable comments. We added the following detailed explanation to the footnote of Table 2: “Meta-regression analysis for continuous variables (total no. of patients, age, female participation rate, and BMI); meta-ANOVA analysis for categorical variables (country, study design, crossover study, underlying disease, forest program education, forest therapist involvement, and forest program time).”

Q4: There is a risk of bias due to the small number of studies. The author needs to report the results of Eger's test in addition to the Funnel plot for publication bias. In addition, it is necessary to add related discussions to the limitations on the risk of research bias.

 Responses: We appreciate your valuable comments. We have already performed Egger's test (Linear regression test) and Begg & Mazumdar's test (Rank correlation test). The p-values obtained suggest evidence of publication bias or small-study effects in this meta-analysis; however, it is not recommended to use the above methods, as only small studies were included in the funnel plot (e.g., two for self-esteem). Following the recommendations of Sterne et al. (2011), a test for funnel plot asymmetry is only conducted if the number of studies is ≥10.

Reviewer 2 Report

A well-written review of the psychological and physiological effects of forest therapy! However, consider the following points. I. This time, psychological and physiological data are processed using a database, but how about data on the forest environment, the hardware that produced these effects? For example, do you have data on average tree height, stand density, tree species, forest slope, etc.? I think we also need such forest data. If such data are missing in each paper, why? Please write about this. 2. Psychological measures such as happiness are derived from subjective evaluations and are considered to be greatly influenced by the subject's personality. Don't you think that participating in such a forest experiment shows the subject's active and positive psychology? Please consider this point as well.

Author Response

A well-written review of the psychological and physiological effects of forest therapy! However, consider the following points.

  1. This time, psychological and physiological data are processed using a database, but how about data on the forest environment, the hardware that produced these effects? For example, do you have data on average tree height, stand density, tree species, forest slope, etc.? I think we also need such forest data. If such data are missing in each paper, why? Please write about this.

Responses: We wholly agree with your comment regarding the importance of forest environments; however, the main goal of our meta-analysis was to investigate how a forest visit itself—rather than each factor of a forest environment—affects participants’ psychological and physiological well-being. Early during our research, we briefly considered including factors of forest environments—such as average height, stand density, tree species, and forest slope—in our meta-analysis; however, we soon discovered that very few journals have made a statistically viable connection between each forest environment factor, and changes in subjects’ physiological and psychological parameters. For example, among the 17 studies included in our meta-analysis, only one study—written by Zeng et al. and published in 2020—clearly stated the temperature, relative humidity, radiant heat, noise, etc. of each forest (Table 2). Still, they did not investigate how each forest’s environmental factors influenced subjects’ psychological and physiological parameters; thus, a meta-analysis was not possible due to the lack of detailed data regarding the forest environment in most studies. The environment of the forest as hardware for health effects was therefore expressed as "the forest" in this paper.

  1. Psychological measures such as happiness are derived from subjective evaluations and are considered to be greatly influenced by the subject's personality. Don't you think that participating in such a forest experiment shows the subject's active and positive psychology? Please consider this point as well. 

Responses: We agree with your opinion and have therefore revised the text and added a sentence to the Discussion section (lines 340~352).

“In some studies, the subjects decided whether to participate in the intervention or control group and selected the places for the intervention based on accessibility and preference, leading to selection bias.”

Revised to: “In some studies, subjects decided whether to participate in the intervention or control group, and selected the location for the intervention based on their accessibility and preference. This selection bias implies that the personality of participants may have compromised the interpretation of our meta-analysis.”

Added sentence:

“Furthermore, evidence regarding how the forest-based intervention affects elements of sentiments—such as self-esteem, concentration, and sociability—is lacking. This prompts further research on the relationship between diverse aspects of the subjective sense of well-being and exposure to forests, especially considering the subject's personality.”

## Additional Sentence needs to be added on the Author Affiliation section.

We forgot to mention that two authors (Sung Ryul Shim1 and JinKyung Chang2) equally contributed to writing and publishing this paper; thus, we have added the following sentence to the Author Affiliation section, marking JinKyung Chang as the co-first author: “These authors contributed equally to this work.”
